# Effects of an Indoor Cycling Program on Cardiometabolic Factors in Women with Obesity vs. Normal Body Weight

**DOI:** 10.3390/ijerph17238718

**Published:** 2020-11-24

**Authors:** Marzena Ratajczak, Damian Skrypnik, Piotr Krutki, Joanna Karolkiewicz

**Affiliations:** 1Department of Biology and Anatomy, Poznan University of Physical Education, 61-871 Poznan, Poland; krutki@awf.poznan.pl; 2Department of Treatment of Obesity, Metabolic Disorders and Clinical Dietetics, Poznan University of Medical Sciences, 61-701 Poznan, Poland; damian.skrypnik@gmail.com; 3Department of Food and Nutrition, Poznan University of Physical Education, 61-871 Poznan, Poland; karolkiewicz@awf.poznan.pl

**Keywords:** indoor cycling, high-intensity interval training, obesity, cardiovascular diseases risk factors

## Abstract

The study aimed to provide evidence on the impact of indoor cycling (IC) in reducing cardiometabolic risk factors. The study compares the effects of a 3 month IC program involving three 55 min sessions per week on women aged 40–60 years, with obesity (OW, *n* = 18) vs. women with normal body weight (NW, *n* = 8). At baseline and at the end of the study, anthropometric parameters, oxygen uptake (VO_2_ peak), and serum parameters: glucose, total cholesterol (TC), high-density lipoprotein cholesterol (HDL-C), low-density lipoprotein cholesterol (LDL-C), and triglycerides (TG), insulin, human anti-oxidized low-density lipoprotein antibody (OLAb), total blood antioxidant capacity (TAC), thiobarbituric acid reactive substances (TBARS), endothelial nitric oxide synthase (eNOS), C-reactive protein (CRP), lipid accumulation product (LAP), and homeostasis model assessment of insulin resistance index (HOMA IR) were determined. Before the intervention, VO_2_ peak and HDL-C levels were significantly lower and levels of TG, LAP, insulin, HOMA-IR, and CRP were significantly higher in the OW group compared to those in the NW group. After the intervention, only the OW group saw a decrease in body mass, total cholesterol, OLAb, TBARS, and CRP concentration and an increase in total body skeletal muscle mass and HDL-C concentration. In response to the IC training, measured indicators in the OW group were seen to approach the recommended values, but all between-group differences remained significant. Our results demonstrate that IC shows promise for reducing cardiometabolic risk factors, especially dyslipidemia. After 12 weeks of regular IC, the metabolic function of the OW group adapted in many aspects to be more like that of the NW group.

## 1. Introduction

Prescribing physical activity to people with obesity is no longer a rarity and in fact has become recognized as a necessity. Specifically, moderate-intensity aerobic training is among the most highly recommended measures for people with obesity because it is well tolerated and favorably affects cardiometabolic risk factors, including dyslipidemia and type 2 diabetes [1]. Other forms of exercise—for example, high-intensity anaerobic exercises or exercises that involve a high dose of strength training—are not often prescribed to people with obesity given concerns over effectiveness as well as patient safety [2].

However, recent studies indicate that endurance training that incorporates small bouts of anaerobic exercise can be more beneficial for people with obesity than moderate-intensity training [3], especially at a comparable level of energy expenditure [4]. Furthermore, a large body of evidence indicates that strength training is just as important for the reduction of cardiometabolic disorders in people with obesity as exercises that increase cardiovascular and muscle endurance [1].

A growing number of studies indicate that the diversity of physical training—understood as the combination of exercises of different intensities and modalities in the same training session—strengthens its effectiveness in reducing cardiovascular risk factors [5,6,7]. Composing a training session of different exercises of varying intensity and purpose (strength, endurance, etc.) can be a good strategy for increasing energy expenditure [8]. Furthermore, combining exercises is likely to result in a greater reduction of myocyte fat content, a greater ability to oxidize fatty acids, increased blood lipid clearance, and increased glucose uptake in muscles [6]. 

Indoor cycling (IC)—also known as spinning—is an increasingly popular but insufficiently studied form of physical activity. IC sessions consist of workout intervals with variable intensity and moderate-to-high involvement of the cardiovascular system as well as the skeletal muscles [9]. IC training sessions can involve aerobic and anaerobic components in various proportions and can be easily modified by changes in pace and changes in the resistance of the stationary bicycle’s flywheel. Indoor cycling programs usually consist of a variety of cycling techniques, including “climbing” (cycling while standing), “jumping” (alternately sitting and standing for a period of time), and “free-wheeling” (cycling at a very fast pace) [10]. However, standard IC classes are rather similar to high-intensity interval training (HIIT). A typical training session includes a few intervals of high intensity (lasting several minutes) interspersed with longer recovery intervals; an entire session lasts approximately 50 min. An additional advantage of IC training is its strength component. The handlebar of the stationary bicycle allows the user to adopt different body positions, engaging the muscles of the trunk and upper shoulder girdle. This supports endurance and strength training [11], as well as the ability to switch from one energy system to another.

A small literature indicates that IC is an effective intervention in improving aerobic capacity [12] as well as lipid and carbohydrate metabolism [9,13]. However, little is known about the optimal combination of exercise intensity and volume for adaptations of IC training in a clinical setting. To the best of our knowledge, there are no papers in peer-reviewed scientific journals dedicated to the impact of IC on biochemical factors related to endothelial dysfunction or other cardiovascular risk factors in women with obesity. We hypothesize that IC training can mitigate cardiometabolic risk factors in women with obesity and improve examined indicators toward that occurring in women with normal body weight. 

Therefore, in this study, we aimed to (1) evaluate the potential clinical effectiveness of indoor cycling in the treatment of obesity and (2) provide-up-to-date evidence on the impact of indoor cycling in reducing cardiovascular risk factors—namely, dyslipidemia, insulin resistance, and endothelial dysfunction.

## 2. Materials and Methods

### 2.1. Study Design

The study was designed as a prospective exercise intervention trial comparing the effects of a 3 month physical training program on women with obesity (OW) vs. women with normal body weight (NW). Outside the implemented program, all participants were instructed to maintain their normal levels of physical activity and normal dietary patterns (and not to use any dietary supplements). At baseline and at the end of the study, anthropometric parameters and physical capacity were measured and blood samples were collected. Dietary intake was assessed in interviews conducted at baseline and at the end of the trial. The studied women consumed Central European food products, and their consumption of monosaccharides exceeded 10% of the daily caloric intake. Food diary analysis eliminated the significant impact of functional foods on research results. Nutrient intake and total caloric intake during the study were constant in both groups. 

### 2.2. Study Participants

From the 163 registered women with obesity (body mass index (BMI) ≥30 kg/m^2^; waist circumference >80 cm, with stable body weight in the month prior to the trial), screened at the outpatient clinic of the Department of Internal Medicine, Metabolic Disorders, and Hypertension, University of Medical Sciences, Poznań, Poland, a total of 23 met inclusion criteria and accepted the invitation to participate in the study. One of the 23 women resigned from the study during the intervention due to personal reasons. Four participants of extreme age (<41 and >60) were excluded from the analysis. Ultimately, the OW group consisted of 18 women with obesity.

Eight women with normal body weight (BMI ≤24.9 and ≥18.5 kg/m^2^) were enrolled after responding to an advertisement for the study. All women in this group completed the intervention and were included in the statistical analyses.

Exclusion criteria for both study groups included the following: a secondary form of obesity and/or a secondary form of hypertension; diabetes mellitus; a history of coronary artery disease; stroke; congestive heart failure; clinically significant heart arrhythmias or conduction disorders; malignancy; poorly controlled hypertension (systolic blood pressure > 140 mmHg and/or diastolic blood pressure > 90 mmHg) and/or modifications to antihypertensive treatment; lipid disorders requiring the implementation of drug treatment; clinically significant abnormalities in liver, kidney, or thyroid gland function; clinically significant acute or chronic inflammatory process within the respiratory, digestive, or genitourinary tracts or within the oral cavity, pharynx, or paranasal sinuses; connective tissue disease or arthritis; history of infection in the month prior to the study; nicotine, alcohol, or drug abuse; and/or any other condition which, according to the researchers, would indicate that participation in the study may be detrimental to the participant.

### 2.3. Anthropometric Measurements and Body Composition Assessments

Anthropometric measurements were taken with subjects wearing light clothing and no shoes. BMI was calculated as weight divided by the square of height (kg/m^2^). Obesity was defined as BMI ≥ 30 kg/m^2^. Waist circumference (cm) was measured at the level of the iliac crest at the end of normal expiration. Hip circumference was measured at the maximum protuberance of the buttocks. 

Body composition analysis was assessed using dual-energy X-ray absorptiometry (DXA; GE Healthcare Lunar Prodigy Advance, GE Medical Systems, Milan, Italy). Subjects were given complete instructions on the body composition analysis procedure and were instructed not to make any intense physical effort in the 24 h prior to body composition measurement. Total body fat mass (TBFM) was determined using standard scan mode (in the case of normal body weight and moderately obese subjects) or thick scan mode (in the case of extremely obese subjects); the absorbed doses of radiation were 0.4 and 0.8 μGy, respectively. Total body skeletal muscle mass (TBSMM) was calculated from appendicular lean soft tissue (ALST)—evaluated using DXA and the skeletal-muscle-prediction model created by Kim et al. [14]: TBSMM=(1.13×ALST)−(0.02 ×age )+(0.61 ×sex )+0.97 (sex = 0 for female).

### 2.4. Cardiorespiratory Fitness

To determine the subjects’ physical capacity, a Graded Exercise Test (GXT) was performed on an electronically braked cycle ergometer (Kettler DX1 Pro, Kettler, Ense, Germany). The GXT began at a work rate of 25 W (60 rev/min) and was increased by 25 W every two minutes until the subject could no longer maintain the required pedal cadence. Exercise tests lasted 4–14.5 min and were conducted between 8:00 a.m. and 12:00 p.m. During the GXT, expired gases, minute ventilation, and heart rate (HR) were monitored continuously with an automated system (Oxycon Mobile, Viasys Healthcare, Hoechberg, Germany). Peak oxygen consumption (VO_2_) was defined as the highest 15 s averaged VO_2_ obtained during the GXT’s final exercise load. Ventilatory threshold (VT) was determined with computerized regression analysis using the V-slope method on the slopes of CO_2_ output versus O_2_ uptake. VT was expressed as a ventilatory threshold heart rate.

### 2.5. Blood Analysis

Blood samples for biochemical analyses were taken from a basilic vein after overnight 12 h fasting. Serum parameters were measured using commercially available immunoassays. Fasting blood glucose (GLU), total cholesterol (TC), high-density lipoprotein cholesterol (HDL-C), low-density lipoprotein cholesterol (LDL-C), and triglycerides (TG) were assessed using tests manufactured by Siemens Healthcare Diagnostics, Inc. (Malvern, PA, USA). Insulin was measured using the HI-14K Human Insulin-Specific RIA immunoassay (Merck, Germany). Human anti-oxidized low-density lipoprotein antibody (OLAb) was analyzed using an ELISA kit from MyBioSource, Inc. (San Diego, CA, USA). As for plasma parameters, total blood antioxidant capacity (TAC) and thiobarbituric acid reactive substances (TBARS) were measured using tests made by CELL BIOLABS, Inc. (San Diego, CA, USA). The activity of endothelial nitric oxide synthase (eNOS) was evaluated using a kit from MyBioSource, Inc. (San Diego, CA, USA). C-reactive protein concentration (CRP) was measured using a highly sensitive marker from CELL BIOLABS, Inc. (San Diego, CA, USA). Insulin resistance was calculated as follows using a homeostasis model assessment of insulin resistance index (HOMA-IR): fasting insulin (mU/mL) × fasting plasma glucose (mmol/L)/22.5, as described by Matthews et al. [15]. Lipid accumulation product (LAP) was computed using waist circumference and fasting triglyceride level (in mmol/L): (WC − 58) × TG [16].

### 2.6. Exercise Training Protocol

The study’s 3 month intervention consisted of a physical exercise program involving three indoor cycling sessions per week, for a total of 36 training sessions. Subjects exercised on Schwinn Evolution cycle ergometers (Schwinn Bicycle Company, Boulder, CO, USA). Each session lasted approximately 55 min. Training sessions consisted of a 5 min low-intensity warm-up (50–65% of maximum heart rate (HRmax)), a 40 min main training block (65–95% of HRmax), 5 min of non-weight-bearing cycling, and, finally, a 5 min low-intensity cool-down with stretching and breathing exercises (Figure 1). The main training block involved three to four 4 min high-intensity intervals (exceeding 80% of HRmax, often reaching anaerobic threshold) and was interspersed with recovery periods (65–80% of HRmax). 

Indoor cycling sessions resemble riding a bike outdoors on different terrains. A flat, seated position was used during warm-ups and cool-downs, whereas standing flat, jumps, seated climbs, seated flat sprints, and standing climb movements were applied during high-intensity intervals (though there was not a defined pattern to each training session). Movements corresponded to a compilation of a music to a large extent. Static stretching was performed in a standing position, with support or thrust on the bike.

During IC sessions, HR was monitored with a Suunto Fitness Solution device (Suunto, Vantaa, Finland). To ensure that the intended exercise intensities were obtained, we monitored the average percent of the HRmax during the entire training session (Figure 1). The mean heart rate for all participants during the entire intervention was 81.5% of HRmax. Analysis of peak HR values recorded during the sessions indicated that all participants’ average peak HR eventually exceeded the post-training values of ventilatory threshold HR, indicating a switch from aerobic to anaerobic exercise. This confirms the high intensity of the exercise intervals. Training sessions were conducted between February to April. Mean attendance at the IC sessions was 73% for the NW group and 83% for the OW group.

### 2.7. Statistical Analyses

All data are expressed as mean ± standard deviation (SD). All statistical analyses were performed using the STATISTICA 13.3 software package (TIBCO Software, Inc., Palo Alto, CA, USA). Some data violated normality and demonstrated heterogeneous variability; therefore, with the exception of the Student’s *t*-test, non-parametric tests were used for all analyses. The Mann–Whitney U test was used to evaluate statistically significant differences between groups. The Wilcoxon rank-sum test was used to assess statistically significant differences between variables before and after the 3 month intervention. Spearman’s rank analysis and quadratic equations were used to calculate correlation coefficients between maximum VO_2_ peak and other variables. A sample size was determined according to changes in VO_2_ peak. A total of six subjects in the OW group and six subjects in the NW group were calculated to yield at least 80% power for detecting an intervention effect at the 0.05 α level, with a detectable effect size of 0.8.

### 2.8. Ethical Considerations

Informed consent was obtained from all participants, and the study was approved by the Ethics Committee of Poznan University of Medical Sciences (case no. 1077/12; supplement no. 753/13). The study conformed to all ethical issues included in the Helsinki Declaration. The study was retrospectively registered on ClinicalTrials.gov on July 2, 2020 under the ID NCT04456192. The study protocol is available at https://clinicaltrials.gov/ct2/show/NCT04456192.

## 3. Results

At the study entry, both groups, women with normal body weight (NW) and women with obesity (OW), had similar age and height. Compared to the NW group, the OW group had significantly higher mean total body mass (95.22 vs. 66.06 kg, *p* < 0.001) and significantly higher mean BMI (36.2 vs. 24.3 kg/m^2^, *p* < 0.001) (Table 1). 

Table 2 shows average values and standard deviations of anthropometric and body composition measures and VO_2_ peak in both the OW and NW group before and after the intervention. As expected, almost all baseline measures were greater in the OW group compared to those in the NW group. In response to IC training, more changes in anthropometric, body composition, and physiological parameters were observed in women with obesity compared to those in women with normal body weight.

Total body skeletal muscle mass was the only parameter that was not different between the OW and NW groups before intervention. After the intervention, only women with obesity saw a decrease in body mass and an increase in TBSMM. In response to the IC training, anthropometric and body composition indicators except TBSMM, decreased in the OW, but remained higher compared to those in the NW. VO_2_ peak increased in both the NW and OW groups but was still higher in the NW group compared to that in the OW group after the intervention. In relation to changes observed in response to the intervention, the only difference between groups was a greater reduction (∆) in hip circumference in the OW group compared with that in the NW group. 

Table 3 shows the biochemical variables associated with cardiometabolic risk and endothelial dysfunction among the study participants before and after IC training. Both the NW and OW groups had similar baseline levels of TC, LDL-C, OLAb, glucose, TBARS, TAC, and eNOS activity. Before the intervention, HDL-C levels were significantly lower and levels of TG, LAP, insulin, HOMA-IR, CRP were significantly higher in the OW group compared to those in the NW group. 

Different changes were observed in women with obesity and women with normal weight after the IC training. In the NW group, the only significant change was an increase in mean TG levels. In the OW group, there was a significant increase in mean HDL-C levels and a significant decrease in mean TC, OLAb, TBARS, and CRP; mean TG and LDL-C levels remained similar. 

Furthermore, although there was no difference in TAC levels between the NW and OW groups before the intervention, TAC concentration was significantly different between the groups after the intervention (although TAC levels did not change significantly in either study group during the intervention). Nevertheless, in both the NW and OW groups, TAC concentration remained within the reference range of 1.10–1.54 mmolCRE·L^−1^ (Cell Biolabs, Inc.), typical for the European population. We additionally detected a significant intergroup difference in changes (∆) in OLAb concentration: Antibody levels decreased in the OW group and increased (though not significantly) in the NW group.

Though mean TBARS concentration has not changed in the NW group, we detected a nonlinear squared correlation between VO_2_ peak and TBARS concentration in normal-weight women after the IC intervention. For women with a VO_2_ peak up to 24.5 mL/(kg×min), the higher the oxygen uptake the higher the concentration of TBARS. In contrast, in women with higher VO_2_ peak values, higher the oxygen uptake levels corresponded to lower concentrations of TBARS (Figure 2).

We detected an interesting nonlinear squared correlation between VO_2_ peak level CRP concentration in women with obesity after the intervention. In women with lower oxygen uptake [up to 20 mL/(kg×min)], CRP concentration fell as oxygen uptake got higher, whereas in women with higher VO_2_ peak values [more than 20 mL/(kg×min)] CRP concentration increased as oxygen uptake got higher (Figure 3A). In the same group, it was observed that the greater the increase in VO_2_ peak, the lower the change in CRP (Figure 3B).

## 4. Discussion

The main motivation for many people with overweight or obesity to begin a physical activity is weight control rather than the improvement of their cardiovascular fitness [17]. Our results show that 12 weeks of regular indoor cycling can be a potent stimulus for improvements in anthropometric and body composition parameters as well as VO_2_ peak. As expected, these changes were more noticeable in women with obesity compared to changes in women with normal body weight. A significant decrease in waist circumference, hip circumference, BMI, and total body fat mass were observed in both groups, but only women with obesity also experienced a decrease in body weight (Table 2). These results suggest that in order to reduce body weight, different training programs are needed for women with obesity vs. those with normal body weight. Significant benefits of the IC training intervention for women with obesity included an increase in muscle mass and an improvement in oxygen metabolism, as evidenced by an increase in VO_2_ peak. Mechanisms underlying the increase of VO_2_ peak with indoor cycling are thought to involve increased ATP generation during high-intensity intervals (via phosphocreatine degradation and muscle glycogenolysis), improvement in vascular function, an increase in cardiac output, and an increased muscle oxidative capacity [18]. Therefore, we suspect that improvements in body weight, waist and hip circumferences, and body composition after indoor cycling were likely the result of an upregulation of bioenergetic oxidation and thus an increase in fat oxidation. In response to the IC training, we observed an increase of total body skeletal muscle mass and a greater reduction in hip circumference in the OW group (Table 2). Thus, we suspect that anabolic signaling mechanisms in the muscles of women with obesity may differ from those of normal-weight women under the influence of the same physical training program. Perhaps a higher training volume threshold is required to increase muscle protein synthesis in normal-weight women. To this end, dysregulated signaling of mammalian target of rapamycin complex 1 (mTORC1) has been observed in the muscles of obese vs. normal-weight adults [19]. In particular, muscles of individuals with obesity show an increased mTORC1 concentration and increased mTORC1 phosphorylation compared to the muscles of normal-weight adults [20]. 

Blood-based biomarkers such as fasting blood glucose, total cholesterol, triglycerides, LDL-C, HDL-C, and HOMA-IR are well established indicators of cardiometabolic risk [21]. Although IC resembles HIIT, it may have a different impact on cardiometabolic factors. In our study, before the intervention, women with obesity had significantly lower levels of HDL-C and significantly higher TG levels compared to normal-weight women (Table 3). This is consistent with the well-known finding that elevated BMI is associated with disturbances in lipid metabolism [22]. 

After the IC training, normal-weight women experienced an increase in TG concentration. However, it should be noted that the TG concentration remained within the recommended value for European women (TG ≥ 1.6 mmol/L) both before and after the intervention. Other HIIT interventions have also resulted in an increase in TG concentrations without any changes in lipid profile, most notably a trial comparing adolescents with normal weight vs. obesity: After a 7 week HIIT program, no changes in lipid profile were found except for an increase in triglycerides concentration [23], similarly after 12 weeks [24]. The authors suggest that lowering of triglyceride levels may only be apparent in those who present with abnormal pre-intervention triglyceride profiles [23]. 

In the OW group, we observed a significant increase in HDL-C concentration and a simultaneous decrease in TC, and human anti-oxidized low-density lipoprotein (oxLDL) antibody (OLAb) (Table 3). It should be mentioned that exercise can increase oxidized HDL (oxHDL) and the ratio of oxHDL to HDL while decreasing oxLDL and TBARS concentrations. Furthermore, exercise can affect HDL-C function, including the promotion of the reverse cholesterol transport and lipid peroxide transport clearing [25,26]. The reason for the lack of change in TG concentration in response to IC training among women with obesity is unclear, but it could be the consequence of the greater variation of TG concentrations in the OW group; alternatively, it is possible that a longer intervention period is needed to lower plasma TG levels. With respect to blood lipids, a meta-analysis of 51 studies involving exercise interventions of moderate-to-high aerobic intensity (some with a dietary intervention) showed a large variability in lipid profile: improvements in HDL-C were observed in approximately half of the studies, and reductions in triglycerides, total cholesterol, and LDL-C were observed less frequently [9,27]. All in all, our observations suggest that regular indoor cycling is an effective activity for general lipid profile improvement in women with obesity.

Our results further indicated that cardiovascular risk parameters were related to the occurrence of an abnormal LAP index, for which the cutoff point was 34.2 cm × mmol/L, above which, the risk of metabolic syndrome is increased [28]. In line with previous literature [29], our results indicate a significantly greater LAP index in the OW group compared to that in the NW group both before and after the intervention, indicating an unhealthy metabolic phenotype for women with obesity. However, women with obesity showed a decrease in LAP in response to IC training (Table 3).

Particularly, perivascular adipose tissue from obese individuals seems to promote local inflammation and impairment of endothelial function, thus providing a link between adipose tissue and vascular disease [30]. As expected, our results indicate a significantly higher baseline level of the inflammatory marker CRP in women with obesity compared to those with normal weight. It has been demonstrated that obesity is related to CRP levels and that adipose tissue likely modulates CRP levels [31]. IC training caused a significant decrease in CRP concentration in women with obesity, but CRP levels in this group were still higher than in normal-weight women (Table 3). IC training could decrease CRP levels through several mechanisms. The main mechanism through which physical activity can modulate levels of inflammation is via muscle-derived cytokines (myokines) released by the contraction of skeletal muscles. As with other comprehensive forms of physical activity, it is likely that the benefits of IC training are due to the use of considerably high muscle mass during exercise [32], which further increases the production of myokines.

It has been suggested that decreases in CRP levels may be correlated with an increase in cardiorespiratory fitness [33]. Our findings are consistent with this, but only in women with obesity (who had higher initial values of CRP) and only in women with oxygen uptake up to 20 mL/(kg×min) (Figure 3). In women with higher VO_2_ peak values [>20 mL/(kg×min)] CRP concentration increased with higher oxygen uptake after the IC training (Figure 3). The observation that CRP concentration did not change in the NW group is consistent with other studies demonstrating that in non-obese people concomitant weight loss is required before a decrease in CRP concentrations becomes apparent [34].

Growing scientific evidence now supports the role of several non-traditional risk factors as potential modulators of the endothelial phenotype in obesity; these include endothelial nitric oxide synthase (eNOS), total blood antioxidant capacity (TAC), and the oxidative stress marker TBARS [35]. Differences in the bioavailability of nitric oxide (NO) are particularly relevant for the prevention of cardiometabolic disorders [36]. The availability of NO in the plasma can be used as a measure of endothelial dysfunction but, given NO’s short half-life, it is more common to use the activity of eNOS as a proxy instead [37]. Exercise involving large muscle groups causes endothelium-dependent improvements in vasodilatation throughout the entire body. Adaptive changes in endothelial function have been shown to be associated with physical training, depending (to a large extent) on the activity of eNOS and changes in oxidant/antioxidant balance [38]. The present study reported similar baseline levels of TAS, TBARS, and activity of eNOS in both obese and normal-weight women. However, after the IC training program, a reduction in TBARS concentration was observed only in the OW group (Table 3). Although no significant changes in post-training TBARS concentration were observed in the normal-weight women, we observed an interesting nonlinear squared correlation between TBARS and VO_2_ peak in this group: TBAR concentration increased with higher oxygen uptake for women with a VO_2_ peak up to 24.5 mL/(kg×min), but decreased with higher oxygen uptake for women with a VO_2_ peak > 24.5 mL/(kg×min). Such a relationship may indicate that women with a higher VO_2_ peak can better tolerate a homeostatic disturbance during physical activity, as indicated by the lower levels of TBARS production (Figure 2).

Many recent studies confirm that the endothelium is the target organ of insulin; under conditions of insulin resistance, the microvascular activity of insulin is disturbed, which significantly contributes to the development of cardiovascular disease [39]. In addition, the HOMA-IR index, a measure of insulin resistance, has been shown to predict cardiovascular disease in Caucasian individuals from the general population [40]. Prior to the IC training program, women with obesity had significantly higher levels of insulin and HOMA-IR index compared to normal-weight women. These observations confirm that the reduction of insulin sensitivity in people with obesity is compensated for by hyperinsulinemia, which means that some people with insulin resistance do not develop type 2 diabetes for many years. Studies indicate that regular physical training leads to beneficial physiological changes, in particular, improvements in the insulin sensitivity of the skeletal muscles and liver as well as improvements in glucose uptake and utilization by skeletal muscles [41]. Exercise-induced changes in skeletal muscle cells are strongly associated with an increase in insulin sensitivity of both muscle tissue itself and other tissues [42]. Our study’s IC intervention did not induce any changes in HOMA-IR, although it did result in modest improvements in muscle mass in the OW group (Table 3). Our findings seem to be inconsistent with those of previous studies demonstrating that high-intensity training was accompanied by improved glycemic control in people with obesity and/or type 2 diabetes [43,44]. On the other hand, it should be recognized that the HOMA-IR assessment is not always sensitive to small but significant changes in insulin sensitivity [45].

It is possible for people with normal body weight to develop features of metabolic syndrome. However, normal-weight women in our study met the strict exclusion criteria, which means that they were generally healthy, and the training program did not significantly improve their metabolic function. We hypothesize that this intensive, interval indoor cycling protocol could be just as beneficial for women with normal BMI and cardiometabolic disorders as it was for the women we studied with obesity. 

Limitations of our study include a relatively small sample size and a wide age range of participants. Although the small sample size did not prevent us from seeing clear differences between groups, our results cannot be generalized to other populations. Additionally, body composition influences the performance of the training program, for example, people with obesity usually achieve lower VO_2_ peak increases, in that way, conclusions from studies comparing people with different fat/muscle mass should be made with caution. The advantages of this study include directly supervised exercise sessions and controlled diets for all participants. We recommend further studies using a larger sample and a more homogenous group.

## 5. Conclusions

In conclusion, this study revealed that 12 weeks of regular indoor cycling is a potent stimulus for improvements in body composition and aerobic capacity in women with normal weight and obesity, but it only reduced body weight and increased muscle mass in women with obesity. Indoor cycling further improved lipid metabolism, especially in women with obesity, and reduced levels of the inflammatory marker CRP. Overall, our results demonstrate that indoor cycling shows promise for reducing cardiometabolic risk factors, especially dyslipidemia, in women with obesity. After 12 weeks of regular indoor cycling, the metabolic function of women with obesity adapted in many aspects to be more like women with normal body weight.

## Figures and Tables

**Figure 1 ijerph-17-08718-f001:**
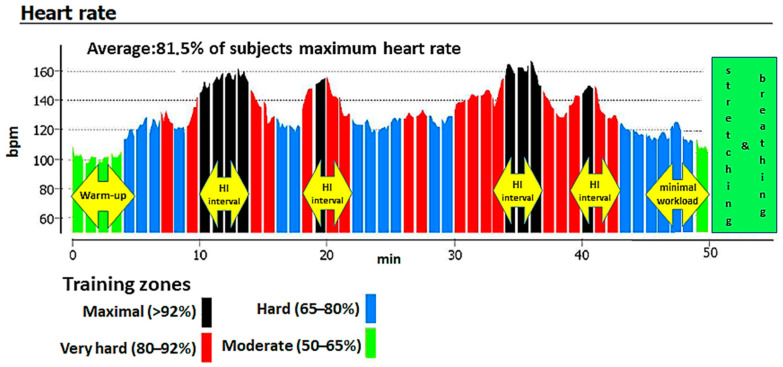
An example indoor cycling (IC) training session based on a heart rate graph from a Suunto Fitness Solution device.

**Figure 2 ijerph-17-08718-f002:**
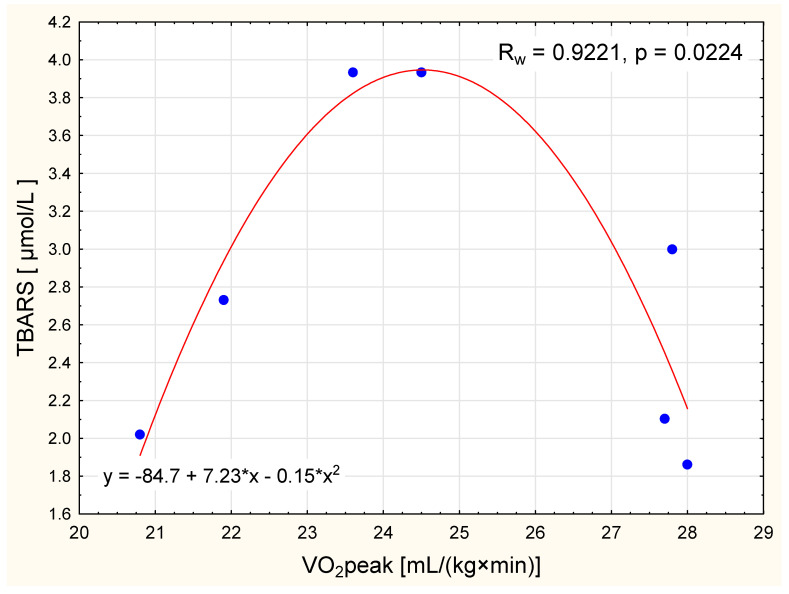
The relationship between VO_2_ peak and level of thiobarbituric acid reactive substances (TBARS) in normal weight women after the intervention (nonlinear squared correlation).

**Figure 3 ijerph-17-08718-f003:**
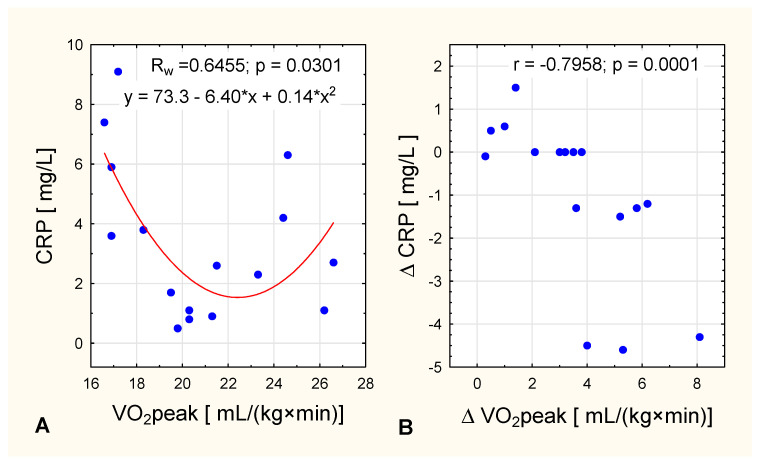
In women with obesity, the relationship between VO_2_ peak and (**A**) post-intervention CRP concentration (nonlinear squared correlation) and (**B**) and change in CRP concentration across the intervention.

**Table 1 ijerph-17-08718-t001:** Characteristics of the study groups before the intervention.

Variables	Normal Weight *n* = 8	With Obesity *n* = 18	*p*
Age (years)	47 ± 5.04	51 ± 6.66	0.167
Height (m)	1.65 ± 0.07	1.62 ± 0.06	0.237
Body mass (kg)	66.06 ± 5.48	95.22 ± 19.6	**0.000**
BMI (kg/m^2^)	24.3 ± 1.31	36.2 ± 5.64	**0.000**

Data are presented as mean ± SD. BMI, body mass index. *p* values below 0.05 are bolded.

**Table 2 ijerph-17-08718-t002:** Anthropometric, body composition, and physiological parameters in the study participants.

	Before Training			After Training					
	Normal Weight Women *n* = 8	Women with Obesity *n* = 18	NW vs. OW		Normal Weight *n* = 8		With Obesity *n* = 18	NW vs. OW	∆NW vs. ∆OW
				Post-Training vs. Baseline		Post-Training vs. Baseline			
	$\bar{X}\pm \mathrm{SD}$	$\bar{X\;}\pm \mathrm{SD}$	*p*	$\bar{X}\pm \mathrm{SD}$	*p*	$\bar{X}\pm \mathrm{SD}$	*p*	*p*	*p*
Body mass (kg)	66.06 ± 5.48	95.22 ± 19.6	**0.000**	64.63 ± 4.35	0.093	92.98 ± 19.05	**0.001**	**0.000**	0.343
BMI (kg/m^2^)	24.3 ± 1.31	36.2 ± 5.64	**0.000**	23.79 ± 1.21	0.050	35.36 ± 5.52	**0.001**	**0.000**	0.275
Waist circumference (cm)	85.06 ± 5.7	111.67 ± 11.5	**0.000**	82.06 ± 4.3	**0.050**	106.42 ± 12.14	**0.000**	**0.000**	0.145
Hip circumference (cm)	99.13 ± 3.16	117.89 ± 12.61	**0.000**	97.5 ± 3.3	**0.039**	114.31 ±12.67	**0.000**	**0.000**	**0.043**
TBFM (%)	34.46 ± 4.54	47.49 ± 3.53	**0.000**	33.44 ± 5	**0.030**	45.64 ± 3.51	**0.000**	**0.000**	0.927
TBSMM (kg)	19.89 ± 1.98	21.71 ± 2.95	0.149	20.25 ± 1.93	0.462	22.27 ± 3.49	**0.038**	**0.141**	0.422
VO_2_ peak [mL/(kg×min)]	24.73 ± 2.81	16.93 ± 2.43	**0.000**	29.26 ± 5.22	**0.002**	20.59 ± 3.41	**0.001**	**0.000**	0.254
VT heart rate (bpm)	144.63 ± 12.82	129.56 ± 16.16	**0.029**	146 ± 9.23	0.673	133.65 ± 14.51	0.334	**0.038**	0.977

Data are presented as mean ± SD. NW, normal weight women, OW, women with obesity; BMI, body mass index; TBFM, total body fat mass; TBSMM, total body skeletal muscle mass; VO_2_ peak, peak oxygen uptake; VT heart rate, ventilatory threshold heart rate. *p* values below 0.05 are bolded.

**Table 3 ijerph-17-08718-t003:** Biochemical parameters in the study participants.

	Before Training			After Training					
	Normal Weight Women *n* = 8	Women with Obesity *n* = 18	NW vs. OW		Normal Weight *n* = 8		With Obesity *n* = 18	NW vs. OW	∆NW vs. ∆OW
				Post-Training vs. Baseline		Post-Training vs. Baseline			
	$\bar{X}\pm \mathrm{SD}$	$\bar{X}\;\pm \;\mathrm{SD}$	*p*	$\bar{X}\;\pm \;\mathrm{SD}$	*p*	$\bar{X\;}\pm \;\mathrm{SD}$	*p*	*p*	*p*
TC (mmol/L)	5.84 ± 0.91	5.75 ± 1.05	0.935	5.68 ± 0.5	0.401	5.36 ± 0.95	**0.035**	0.375	0.261
HDL-C (mmol/L)	1.8 ± 0.26	1.33 ± 0.39	**0.005**	1.82 ± 0.31	0.805	1.46 ± 0.37	**0.006**	**0.026**	**0.000**
LDL-C (mmol/L)	3.32 ± 0.61	3.45 ± 0.85	0.708	3.27 ± 0.48	0.754	3.18 ± 0.90	0.139	0.780	0.397
TG (mmol/L)	0.77 ± 0.24	1.55 ± 0.64	**0.000**	0.91 ± 0.25	**0.097**	1.6 ± 0.78	0.677	**0.004**	0.054
OLAb (U/L)*	10.02 ± 4.93	13.95 ± 14.7	0.628	15.11 ± 13.35	0.278	9.34 ± 13.5	**0.008**	0.066	**0.031**
LAP (cm×mmol/L)	20.92 ± 7.66	84.07 ± 38.9	**0.000**	21.73 ± 6.28	0.724	79.95 ± 54.36	0.589	**0.000**	0.534
GLUCOSE (mmol/L)	4.56 ± 0.39	4.67 ± 0.61	0.656	4.67 ± 0.60	0.766	4.89 ± 0.61	0.260	0.644	0.674
INSULIN (µLU/mL)	5.75 ± 2.95	15.32 ± 7.24	**0.000**	8.15 ± 3.91	0.120	15.57 ± 4.71	0.887	**0.001**	0.435
HOMA IR	1.19 ± 0.7	3.25 ± 1.73	**0.000**	1.67 ± 0.84	0.212	3.34 ± 0.99	0.832	**0.000**	0.582
TBARS (μmol/L)	2.66 ± 0.89	3.05 ± 0.79	0.284	2.13 ± 0.46	0.112	2.46 ± 0.67	**0.016**	0.226	0.884
TAC (mmolCRE·L^−1^)	1.15 ± 0.23	1.33 ± 0.22	0.086	1.03 ± 0.23	0.208	1.35 ± 0.19	0.723	**0.000**	0.190
eNOS (ng/mL)	20.98 ± 18.27	22.26 ± 23.74	0.440	24.21 ± 20.32	0.123	26.13 ± 20.41	0.098	1.000	0.977
CRP (mg/L)	1.2 ± 1.24	4.39 ± 2.64	**0.006**	1.21 ± 1.19	0.463	3.37 ± 2.60	**0.049**	**0.011**	0.492

* Detection range 100–1.56 U/L. Data are presented as mean ± SD. NW, normal weight women; OW, women with obesity; TC, total cholesterol; HDL-C, HDL-cholesterol; LDL-C, LDL-cholesterol; TG, triglycerides; OLAb, anti-oxidized low density lipoprotein antibody; LAP, lipid accumulation product; HOMA-IR, homeostasis model assessment of insulin resistance; TBARS, thiobarbituric acid reactive substances; TAC, total antioxidant capacity; eNOS, endothelial nitric synthase; CRP, C-reactive protein. *p* values below 0.05 are bolded.

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
