# Peer review of "Effects of an Indoor Cycling Program on Cardiometabolic Factors in Women with Obesity vs. Normal Body Weight"

_ijerph, 2020, doi:10.3390/ijerph17238718_

Round 1
Reviewer 1 Report
I read the revised version of the paper. I think that the authors improved very much the manuscript and its contents. The paper is much more fluent.
Reviewer 2 Report
Based on the author's response and resubmitted manuscript, all changes made are accepted. It might also interesting to recruit more participants, but it's too late now, considering the COVID-19 pandemic. The changes presented by the authors have been very positive.
Reviewer 3 Report
All correct, they have taken my suggestions into account, thank you. Nice job.
This manuscript is a resubmission of an earlier submission. The following is a list of the peer review reports and author responses from that submission.
Round 1
Reviewer 1 Report
Obesity is a universal phenomenon. It is not the mere fact that the obese population is increasing in quantity, but the increase in its share in the population structure. In recent years, the intensity of this phenomenon, especially in more developed countries, has increased significantly. A longer human life is a very positive phenomenon, provided that health and fitness are preserved in old age. The increase in the percentage of obese women also poses a challenge to the social policy of the state, as there is a need to build special programmes guaranteeing access to physical exercise and health services that promote healthy ageing.
The authors of the reviewed article focused on seeking and combining new aspects of cardio metabolic factors in women with obesity who practice indoor cycling Program. They have deeply rooted their considerations in properly selected international literature. Congratulations on the idea and I encourage you to further research. However, methods presented in the manuscript are unclear, this must be improved.
Abstract:
- The authors should write about the details of participants (e.g. number of participants, sex, age).
Materials and Methods:
- The authors should add the recruitment process of the participants.
- The number of participants in the Abstract and Methods study is not the same. You have to review it and explain it better, the total number of participants has not been clear to me.
- The reviewer and the potential readers cannot know the study period. It should be added.
- The authors should mention about the sample size calculation of this study.
Results:
- The number of participants in the Tables is not the same. You have to review.
Disscusion
- The limitations of the study are well addressed but it would be good to review them again and add some more.
Reviewer 2 Report
Thank you for inviting me to review the manuscript (ID ijerph-954524) entitled “Effects of an Indoor Cycling Program on Cardio-Metabolic Factors in Women with Obesity vs. Normal Body Weight” reported by Ratajczak and colleagues. The authors evaluated the implications of indoor cycling (IC) practices in reducing cardio-metabolic risk factors. Healthy and obese women participated in this study. Each participant was invited to practice IC 3 times a week for 3 months. Biochemical, inflammatory, and cardiorespiratory markers, anthropometric and body composition parameters were analyzed at the baseline and at the end of the study.
Major point:
This paper has a small number of subjects. In the abstract section, authors reported that 31 women participated in this study. According to the material and methods, eighteen obese and eight normal women were enrolled. Participants excluded based on their conditions or own decision cannot be considered subjects of the study. Also, results reported in table 1 are well determined by literature. Lastly, the significant results reported by the authors are unsafe because the small sample size renders their design insufficiently powered. I suggest reconsidering this work if the authors recruit more subjects.
Minor points:
The quality of tables and figure 1 should be improved.
It’s necessary to review the reference guidelines.
I recommend that the paper is edited by a native English speaker. One very basic example involves the word “comperes”, which should be “compares”.
Reviewer 3 Report
To:
Editorial Board
International Journal of Environmental Research and Public Health
Title: “Effects of an Indoor Cycling Program on Cardio-Metabolic Factors in Women with Obesity vs. Normal Body Weight”
Dear Editor,
I read this manuscript and I think that:
- The mass can impact on the performance of the training program (see also the paper from Meleleo D et al. Eur J Sport Sci. 2017 Jul;17(6):710-719). This should be discussed in a dedicated limitation section. Please provide.
- Table 1 should be updated. ALL the clinical, anthropometric, and laboratory characteristics of the study population should be included and compared. Please update.
- The small sample size is a limitation of the study. Please discuss this point in a dedicated limitation section.
- A post-hoc sample size calculation should be provided.
- Dietary pattern of the patients had not been listed: this can impact on final outcomes.
- Indeed, nutraceuticals – which can be often included in daily foods – can influence cardiovascular risk factors such as hypertension, dyslipidemia, etc. The authors should take into account such a point and discuss it (see for example Scicchitano P et al. Journal of Functional foods 2014;6:11-32).